# Flexible Perfluoropolyethers-Functionalized CNTs-Based UHMWPE Composites: A Study on Hydrogen Evolution, Conductivity and Thermal Stability

**DOI:** 10.3390/ma15196883

**Published:** 2022-10-03

**Authors:** Maurizio Sansotera, Valeria Marona, Piergiorgio Marziani, Nadka Tzankova Dintcheva, Elisabetta Morici, Rossella Arrigo, Gianlorenzo Bussetti, Walter Navarrini, Luca Magagnin

**Affiliations:** 1Dipartimento di Chimica, Materiali e Ingegneria Chimica, Politecnico di Milano, Via Mancinelli 7, 20131 Milan, Italy; 2Consorzio Interuniversitario Nazionale per la Scienza e Tecnologia dei Materiali (UdR-PoliMi), Via G. Giusti, 9, 50121 Firenze, Italy; 3Dipartimento di Ingegneria, Università di Palermo, Viale delle Scienze, Ed. 6, 90128 Palermo, Italy; 4Consorzio Interuniversitario Nazionale per la Scienza e Tecnologia dei Materiali (UdR-Palermo), Via G. Giusti, 9, 50121 Firenze, Italy; 5Advanced Technologies Network (ATeN) Center, Università di Palermo, Viale delle Scienze Ed. 18, 90128 Palermo, Italy; 6Dipartimento di Fisica, Politecnico di Milano, Piazza Leonardo da Vinci 32, 20133 Milan, Italy

**Keywords:** UHMWPE composites, functionalized CNTs, perfluoropolyethers, flexible electrode, hydrogen evolution

## Abstract

Flexible conductive composites based on ultra-high molecular weight polyethylene (UHMWPE) filled with multi-walled carbon nanotubes (CNTs) modified by perfluoropolyethers (PFPEs) were produced. The bonding of PFPE chains, added in 1:1 and 2:1 weight ratios, on CNTs influences the dispersion of nanotubes in the UHMWPE matrix due to the non-polar nature of the polymer, facilitating the formation of nanofillers-rich conductive pathways and improving composites’ electrical conductivity (two to five orders of magnitude more) in comparison to UHMWPE-based nanocomposites obtained with pristine CNTs. Electrochemical atomic force microscopy (EC-AFM) was used to evaluate the morphological changes during cyclic voltammetry (CV). The decrease of the overpotential for hydrogen oxidation peaks in samples containing PFPE-functionalized CNTs and hydrogen production (approximately −1.0 V vs. SHE) suggests that these samples could find application in fuel cell technology as well as in hydrogen storage devices. Carbon black-containing composites were prepared for comparative study with CNTs containing nanocomposites.

## 1. Introduction

Carbon nanotubes (CNTs) have been the object of a great deal of attention since their discovery in 1985 due to their peculiar physico-chemical properties, such as high electrical conductivity, mesoporosity and flexibility [1,2]. Among the high-profile applications devised for this distinctive class of carbon-based nanomaterials, those elaborating on their energy conversion and energy storage capabilities represent a host of promising new technologies viable for the manufacture of supercapacitors [3,4], batteries [5,6], fuel cells [7,8,9], solar cells [10,11], field-effect transistors (FETs) [12], hydrogen storage systems [13,14,15] and in shape memory alloys used for MEMS [16]. In particular, the last two decades have witnessed the development of innovative CNT-based composite materials. The addition of guest species into the flexible and light polymeric matrix allows the transfer of useful mechanical–chemical properties to the host, such as electron mobility [17], electrical conductivity [18,19,20,21] and mechanical resistance [22]. Thanks to these excellent qualities, in addition to low cost and easy processability, conductive polymer nanocomposites are facing new promising industrial applications, in particular as flexible electrodes in energy storage, biomedical mechanical improvements for implants, pressure sensors and interfering electromagnetic shields [23,24,25]. Despite this, the dispersion of carbonaceous fillers in a polymeric medium is difficult to achieve, owing to their propensity to aggregate and possibly segregate at interfaces [26,27,28,29]. This behavior can be counteracted by modifying or otherwise functionalizing the surface of such carbonaceous structures, improving their affinity toward the host matrix [30,31,32]. Notably, direct surface fluorination with elemental fluorine, or functionalization with perfluoropolyether (PFPE) side chains, can be employed to lower surface energy and thus improve the dispersibility of the nanofillers, as well as enhance the chemical stability and processability of the resulting composite materials [33,34,35,36].

In our previous studies, we described an efficient procedure to attain a homogeneous dispersion of PFPE-modified CNTs into a host matrix of ultra-high molecular weight polyethylene (UHMWPE). The functionalization of CNTs and CB using perfluoropolyethers (PFPE), which are well known as highly performing fluorinated polymers suitable for coatings, membranes, solvents and lubricants in harsh environments, can be considered an attractive alternative procedure to fluorination. This procedure, used for different carbon allotropes, consists of the thermolysis of PFPE peroxides to generate carbon-centred perfluorinated free radicals, which bond to the unsaturated moieties on the carbonaceous surfaces of carbon, which act as radical scavengers. Thus, it was expected that the typical properties of fluorinated materials, in particular low surface energy, were transferred to the surface of carbonaceous materials by means of a stable carbon–carbon bonds. The composite was characterized by good viscoelastic properties due to the formation of a percolation network across the UHMWPE matrix at low nanofillers concentration [37]. The nanofillers also showed good electron conductivity [38,39] and slightly higher electrical resistivity compared to bare CNTs and native CB. Indeed, PFPE functionalization allowed the preservation of the polyaromatic bulk structure, maintaining the electrical features of such composites in the typical range of conducting materials. In sharp contrast, surface modification of CNTs and CB with elemental fluorine caused the disruption of the underlying conjugated structure, leading to higher electrical resistivity values compared to pristine fillers and PFPE-containing composites [33,35]. Therefore, the functionalization of carbonaceous structures with PFPE oligomeric chains represents a suitable technique to improve CNTs dispersibility while simultaneously preserving their electrochemical and morphological properties [33,35,37].

In this work, electrical properties, thermal stability and hydrogen storage capabilities of hot compacted composites based on UHMWPE and PFPE-modified CNTs and CB were studied and compared to those of similar composite materials containing pristine fillers.

## 2. Materials and Methods

### 2.1. Materials

The UHMWPE was a commercial grade polymer purchased from Sigma-Aldrich (St. Louis, MO, USA), characterized by a weight-average molecular weight of 3–6 MDa, softening point of T = 136 °C (Vicat, ASTM D 1525B), melting point of T_m_ = 138 °C (determinate by DSC) and a density of 0.94 g cm^−3^ at 25 °C. Multi-walled CNTs produced via chemical vapor deposition (average diameter: 9.5 nm, average length: 1.5 μm, product code: 7000) were acquired from Nanocyl, S.A. (Sambreville, Belgium). Cabot Vulcan^®^ XC72R (Cabot Corporation, Billerica, MA, USA) was used in this work as high conductive and commercially available graphitic CB (particle size 30 nm) [40]. PFPE peroxide (Fomblin^®^ Z PFPE peroxide), prepared by oxidative photopolymerization of tetrafluoroethylene (TFE) [41], was kindly provided by Solvay Specialty Polymers (Brussels, Belgium). Its general chemical structure is as follows: TO(CF_2_CF_2_O)*_m_*(CF_2_O)*_n_*(O)*_v_*T’, where terminal groups T and T’ are trifluoromethyl, CF_3_ or an acyl fluoride group, e.g., C(O)F or CF_2_C(O)F. The specific chemical characteristics of PFPE peroxide are as follows: average molecular weight around 29 kDa, *m*/*n* ratio (between perfluoroethylene oxide, CF_2_CF_2_O and perfluoromethylene oxide, CF_2_O, units) equal to 1.15, peroxidic content of 1.3 wt.% and equivalent molecular weight (EMW) around 1200 g eq^−1^. CF_3_OCFClCF_2_Cl (b.p. 4–41 °C) was used as an inert fluorinated solvent during the chemical treatments of carbonaceous fillers with PFPE peroxide.

### 2.2. Chemical Treatment of Carbonaceous Materials

The functionalization of CNTs and CB with PFPE peroxide was previously accurately studied and documented by Navarrini et al. [33,35]. Modified CNTs specimens were prepared at reagent CNT:PFPE peroxide weight ratios of 2:1 and 1:1, hereafter referred to as CNT_F1_ and CNT_F2_ fillers, respectively. Modified CB specimens, labelled CB_F1_ fillers, were prepared at a reagent CB:PFPE peroxide weight ratio of 1.67:1. Pristine species were labelled as CNT_0_ and CB_0_ fillers.

### 2.3. Preparation of UHMWPE-Based Composites

Composites were produced using all fillers mentioned in Section 2.2, i.e., CNT_0_, CNT_F1_, CNT_F2_, CB_0_ and CB_F1_ [37]. Powdered UHMWPE was mixed with different filler loadings (i.e., 0.5, 1.0, 3.0 and 5.0 wt.%) under magnetic stirring at room temperature until the formation of a homogeneous black powder. The resulting samples were successively hot compacted at 200 °C and 17.2 MPa (2500 psi) for 5 min to produce thin films for characterization purposes. The average thickness of the thin films amounts to 200 ± 40 µm. Due to their polymeric nature and small thickness, these UHMWPE-based nanocomposites show very high flexibility and notable mechanical stability. A series of samples were also prepared at 32.5 MPa (4700 psi). Pure UHMWPE was subjected to the same treatment in order to produce a control sample. Composite specimens were referred to as PE-filler-wt.%, e.g., UHMWPE composite samples containing a 5% weight ratio of MWCNT-based fillers obtained by functionalization with PFPE peroxide at a reagent weight ratio of 2:1 (Section 2.2) were labelled as PE-CNT_F1_-5.

### 2.4. Characterization

The characterization of UHMWPE-based samples comprised of resistivity, voltammetric and thermogravimetric measurements. In addition, UHMWPE-based samples were also observed with different microscopic techniques in order to obtain complementary information on the morphology of UHMWPE-based samples. Scanning Electron Microscopy (SEM) coupled with Energy Dispersive X-ray Spectrometry (EDS) was used for recording the elemental map showing the spatial distribution of fluorine atoms in the cross-section of a sample. Transmission Electron Microscopy (TEM) was employed in order to observe the dispersion of CNTs in the UHMWPE matrix. Atomic Force Microscopy (AFM) was coupled with an electrochemical cell in order to perform electrochemical-AFM (EC-AFM): the AFM head was immersed in an electrolyte, and images were acquired in situ and in real-time during cyclic-voltammetry, showing the morphological changes of a UHMWPE-based sample used as a flexible electrode. Resistivity was estimated using an experimental apparatus purposely designed and realized on the basis of a model described in the literature [42]. A Princeton Applied EG&G 273A (Ametek, Berwin, PA, USA) potentiostat was used to record resistance values by varying the potential difference at the two electrodes between −1.00 V (vs. RE) and +1.00 V (vs. OC) with a scan rate of 20 mV s^−1^, while a piston was applying a pressure of either 17.2 kPa or 32.5 kPa to the sample. The resistivity was then calculated taking into account geometrical parameters such as tube radius, *r*, and sample thickness, *l*, in the following Equation (1),
(1)$$\rho =\frac{R\;\pi \;r^{2}}{l}=\frac{1}{k},$$
where *k* is the specific conductivity, *ρ* is the specific resistivity and *R* is the cylinder resistance.

Experimental settings were optimized towards homogeneous conducting conditions according to the second Ohm’s law. All measurements were repeated three times on each sample in order to obtain averaged values.

Hydrogen evolution and storage capabilities of the composites were assessed by means of voltammetric techniques. The samples, a Pt wire, and a saturated calomel electrode (SCE) were used as working, counter and reference electrodes, respectively, in a common voltammetric three-electrode cell filled with 50 mL of a 1 M H_2_SO_4_ solution, which was purged with nitrogen prior to usage. A Princeton Applied EG&G 273A potentiostat was used for all the experiments. The cyclic voltammetry (CV) experiments were set as follows: three scans starting from +1.0 V, scanning to the vertex potential of −1.5 V and then back to +1.0 V, with reference to SCE. The scan rate was set at 10 mV s^−1^. All potentials in this manuscript are henceforth expressed with respect to the standard hydrogen electrode (SHE). The tests were performed in a typical cell while the overall working conditions of the electrode can be easily associated with energy storage devices of every dimension and shape. The electrode reflects a series of complementary properties related to the flexibility guaranteed by the polymeric matrix, the conductivity of the carbonaceous filler and the stability introduced by the performed PFPE functionalization. These advantages allow us to present our preliminary research on the promising and bright future of these nanocomposites as flexible, innovative electrodes.

A scanning electron microscope (Zeiss EVO–50, Zeiss, Thornwood, NY, USA; working distance 8.0 mm, beam current 100 pA, acceleration voltage 20.00 kV) equipped with an EDS probe (EDS Spectrometer Oxford Inca Energy 200, resolution 132 eV, sensitivity 0.4, accelerating voltage 10 kV, Oxford Instruments, Oxford, UK) was employed for the cross-sectional analysis of the samples. Microscopy was performed on bare samples without deposition of a conductive layer, and the cross-section was obtained by fracturing the sample at liquid nitrogen temperature. TEM was performed using a Philips CM200 FEG electron microscope (acceleration voltage 200 kV) equipped with a Field Emission Gun filament (FEI Company, Thermo Fisher Scientific Group, Eindhoven, NL, USA). A UHMWPE-based sample with dispersed PFPE-modified CNTs was embedded in a low viscosity resin, obtaining a resin block characterized by good adhesion to the fluorinated sample. Ultrathin sections with a thickness of approximately 100 nm were prepared by ultramicrotomy. AFM images were acquired in non-contact mode by a Keysight 5500 (Keysight Technologies, Santa Rosa, CA, USA) system coupled with an electrochemical cell used in a three-electrodes configuration.

Thermogravimetric analyses (TGA) were carried out using a THASS TGA XP-10 (Thass, Friedberg, Germany) analyzer in order to evaluate the thermal stability of the composites. The samples were heated under nitrogen flow from 30 °C to 900 °C on a heating ramp of 10 °C min^−1^.

## 3. Results and Discussion

### 3.1. Preparation of Flexible PFPE-Functionalized CNTs-Based UHMWPE Composites

The flexibility of the UHMWPE functionalized electrodes was investigated with conventional methods of bend and torsional tests. In Figure 1b, the PE-CNT_F2_-5 sample is completely curved, reaching a bending radius of 2 mm, behaving similar to the electrode containing the pristine carbon guest. This test was performed on all the electrodes subjected to the study, and the results were coherent with the electrode displayed. This property coming from the polymeric matrix was not significantly modified by either the addition of carbon-based fillers or their fluorination treatment and did not depend on the degree of fluorination. This important property opens the possibility of the employment of these polymeric composites as electrodes in numerous fields, such as energy storage and pressure sensors [21,22,23,24,25].

### 3.2. Electrical Resistivity

The electrical behavior of UHMWPE-based nanocomposites containing carbon nanotubes and carbon black was studied by resistivity measurements at two different values of applied pressures (Section 2.4). The electrical resistance was measured three times for each sample, and average values, as well as standard deviations, were calculated (see Appendix A). Electrical resistivity values for CNTs- and CB-based nanocomposites are reported in Figure 2 as a function of their composition, expressed in terms of the percentage of filler content.

As expected, composites manufactured using pristine CNTs (PE-CNT_0_) yielded higher values of resistivity at each and every polymer:filler weight ratio compared to PE-CB_0_ composites. The resistivity of the PE-CNT_0_ composites decreased in a seemingly logarithmically fashion, moving from 1.0 to 5.0 wt.% and reaching values comparable to the PE-CB_0_ composites around the maximum polymer:filler weight ratios tested. A certain minimum amount of pure CNT fillers, above 1.0 wt.%, as seen in Figure 2a,b, is thus required for the development of enhanced electrical conductivity features in UHMWPE hot pressed thin films.

Functionalization of carbonaceous fillers has different effects on CNTs and CB. The inherent insulating character of PFPE chains is only evident for CB-based composites, for which functionalization systematically results in a slightly higher resistivity compared to the pristine CB filler. In the case of functionalized CNT fillers, this effect is largely exceeded by morphological effects, also considering the flexibility as perfectly preserved. It was demonstrated that the functionalization of CNTs with PFPE peroxide leads to a slight increase in nanotube resistivity [33]. However, the formation of a three-dimensional CNTs network [43] is strongly facilitated by the presence of PFPE chains, largely overcoming the loss of conductivity due to functionalization.

PE-CNT_F1_ and PE-CNT_F2_ composites show strikingly similar electric behaviors regardless of the polymer:filler weight ratio. As recently reported [37], PFPE chains bond directly to the external walls of CNTs, providing them with a fluorinated liquid-like thin layer characterized by very low surface energy. Its thin layer perfectly fits with the adaptive flexible nature of the nanocomposite. This peculiar feature also allows a homogeneous dispersion of the modified CNTs into the polymeric host matrix. As a result, it is possible for such composite materials to reach saturation conditions, whereas further addition of fillers does not improve conductivity. PFPE-modified fillers per se also suffer from similar issues. There exists a limit to the number of PFPE chains attachable to the carbonaceous particle so increasing the reagent CNT/PFPE peroxide weight ratio past surface saturation does not lead to an increased degree of functionalization. In this instance, TGA measurements on powdered fillers suggest CNT_F1_ samples, i.e., modified CNTs-based fillers produced from the highest starting CNT/PFPE peroxide weight ratio considered herein, already lead to nearly complete surface functionalization. Indeed, the amount of PFPE bonded on CNTs obtained for CNT_F1_ and CNT_F2_ filler samples was 14.3 wt.% and 17.6 wt.%, respectively [33].

Overall, PFPE-modified fillers CNT_F1_ and CNT_F2_ readily impart electrical conductivity to UHMWPE. Resistivity values of composites prepared by dispersing the PFPE-modified CNTs are two to five orders of magnitude lower compared to samples containing unmodified CNTs (Figure 2a,b). This means that fluorinated CNT displays a high surface area and higher electrical conductivity compared to the pristine carbonaceous material (two to five orders of magnitude more), lightweight, perfect hexagonal structure and many unusual mechanical, electrical and chemical properties. Thanks to these properties, a fuel cell containing this hybrid material could easily overcome the actual technological limits, especially related to the high intrinsic resistivity of CNTs [44,45]. Only by increasing the pristine CNTs concentration above 4 wt.% in the UHMWPE matrix is it possible to observe a resistivity decrease comparable to that produced by PFPE-modified CNTs at 1 wt.% concentration. It is important to note that this increase in conductivity is evident at filler loadings as low as 0.5 wt.%, and it is particularly pronounced at loadings in the range of 1 to 3 wt.% (Figure 2a,b). Indeed, PFPE-modified CNTs-based composites feature resistivity values equivalent to or even lower than composites fabricated using commercially available conductive specialty carbon black fillers.

In the literature, it is reported that in the CNT/UHMWPE composite, the pure CNTs were located only at the matrix–grain boundaries and bundles of pure CNTs were observed to be clustered together, constructing a conducting chain between UHMWPE particles [43,46,47,48,49]. TEM was used to observe the details of the structures in the PFPE-modified CNTs-based UHMWPE composites, and these images provided more detailed information on the conductive pathways due to PFPE-functionalized CNTs conserving their morphological advantages, such as flexibility.

Samples obtained at different pressure applied during the sample hot compacting procedure were also compared: 17.2 and 32.5 kPa. The results showed that this pressure only slightly influenced the resistivity trends of the specimens (Figure 2).

### 3.3. Morphology SEM and TEM

The morphological analyses of the PE-CNT_F2_-3 sample were performed using both the Scanning Electrode Microscope (SEM) and the Transmission Electrode Microscope (TEM). The solution used a 1:1 ratio of the CNT-PFPE, and filler loading of the UHMWPE at 3%wt. showed the best resistivity reduction at 17.2 kPa of compacting pressure, so it is able to display the networking condition reached by the functionalized CNT-based nanocomposite.

The elemental map, provided by the EDS probe equipped with the SEM, shows a clear homogenous distribution of the fluorine atoms (brighter spots) inside the polyethylene matrix while the functionalized nanotubes tend to segregate on the surface. This phenomenon can be explained by analyzing the non-polar nature of the UHMWPE matrix. This hydrocarbon is highly hydrophobic, so it drives the fluorinated carbon species towards the border of the section, as underlined in Figure 3b. From these considerations on the cross-section, obtained by fracturing the sample at liquid nitrogen temperature, it is confirmed that during chemical treatment with PFPE peroxide, the polyperoxidic structure of PFPE peroxide allowed the linkage in a network of several connected PFPE chains, gradually growing an entire PFPE layer on the MWCNT surface. This placement explains the increased conductivity of the CNT-based nanocomposite, as already widely demonstrated in Section 3.2. On the other side, CB-based nanocomposites resulted in being less aggregated due to the lower degree of order of the carbon atoms, with smoother effects on the conductivity of the UHMWPE.

TEM images of the PE-CNT_F2_-3 sample are reported in Figure 4 and show the spatial distribution of PFPE-functionalized CNTs inside the UHMWPE matrix.

In Figure 4a, a three-dimensional network of PFPE-functionalized CNTs is clearly shown [43]. The creation of this network was facilitated by the functionalization of CNTs with PFPE chains. The creation of an ordinated network allows the preservation of conductive electrical properties within a flexible polymeric matrix. In fact, the decrease in CNTs surface energy induced by the linkage of perfluorinated chains allowed the dispersion of smaller CNTs agglomerates inside the UHMWPE matrix. Thus, a packed network of CNTs was created, and UHMWPE could reach the conductivity properties with a lower amount of filler. Although the PFPE-functionalized CNTs agglomerates are mainly uniformly dispersed in a network inside the UHMWPE bulk, isolated rows of CNTs aggregates were also observed (Figure 4b). The SEM and TEM morphological analyses can be clearly compared to the bare PE-CNT sample, widely present in state-of-the-art [50,51]. In the cited articles, a non-uniform and non-orientated dispersion of carbon nanotubes inside the UHMWPE matrix is evident proof of the effect of fluorination in the orientated state of CNT-functionalized filaments, as already discussed in these two paragraphs.

### 3.4. Hydrogen Evolution Reaction (HER): Voltammetry and Electrochemical AFM (EC-AFM)

The voltammograms recorded for the CNTs- and CB-based nanocomposites are shown in Figure 5a,b. A marked difference is evident between these two series of measurements. Both measurements present a reduction reaction signal (at higher onset potentials for CB-based composites), but only CNTs-based thin films possess the characteristic peak of hydrogen oxidation. Therefore, CB, unlike CNTs, seems to be unable to store hydrogen electrochemically.

Figure 5a shows the cyclic voltammetry results obtained for PE-CNT_0_-5, PE-CNTF2-5 and PE-CNT_F1_-5 nanocomposites. In particular, during the onward scan, the faradic currents increased due to a reduction reaction, while during the backward scan, as mentioned earlier, an oxidation peak was observed [52,53]. The former is due to proton reduction to produce hydrogen, which is trapped inside nanotubes or within the PFPE liquid-like phase. The latter is due to the oxidation of said absorbed hydrogen [54].

Because excess hydrogen is not trapped but instead released, the faradic current in the corresponding backward scan leads to the oxidation peak in a seemingly diffusion-controlled fashion. Once the absorbed hydrogen is depleted, the peak is reached, and the current rapidly plummets to zero.

In the experimental trials, a strong increment of the faradic currents in the cathodic region was observed in association with the increment of the PFPE content of the fillers, and the expected intense increment of the associated oxidation peak was indeed detected; furthermore, these oxidation peaks shifted towards lower potentials. This effect was presumably promoted by the gas-permeable PFPE coating on the carbon nanotubes, which promoted hydrogen diffusion to and from the CNTs. Therefore, the overpotential decrease and the cathodic current increase can be ascribed to the presence of the PFPE layer that quickly removes hydrogen from the electrically active surface of CNTs, lowering the local concentration of hydrogen and acting as a gas reservoir as well as gas diffusion layer [34,55].

The presence of the PFPE layer on the outer surface of CNTs can also decrease the composite wettability, as well as the diffusion of the polar species (e.g., water and protonated water) toward the CNTs’ surface [37]. With reference to the experimental data presented, it is possible to notice that the diffusion-enhancing effect of the PFPE layer is dominant over the hydrophobic one.

In fact, hydrogen oxidation current peaks are systematically more pronounced with both increasing PFPE content in the filler (Figure 5a) and increasing filler load at the same PFPE content (Figure 6a).

Furthermore, PFPE content and filler load influence the hydrogen oxidation overpotentials obtained during backward scans in a similar manner. The overpotential associated with hydrogen oxidation reactions (Figure 5a and Figure 6a) decreases with increasing PFPE and filler content.

Performing multiple consecutive cyclic voltammetry scans shows that the hydrogen oxidation peak for all tested samples (Figure 6b reports PE-CNT_F2_-3 results as an example) is systematically observed around 1.0 ± 0.1 V (vs. SHE). This feature suggests that it is steadily possible for the electrode to evolve and store hydrogen again during the onward scan.

The decrease of the overpotential for hydrogen oxidation in samples containing PFPE-functionalized CNTs suggests that the latter could find application in fuel cell technology, in particular as catalyst support in the anodic compartment of polymer exchange membrane fuel cells (PEMFCs), where the hydrogen oxidation reaction takes place (HOR). Indeed, the utilization of CNTs in PEMFC devices has been proposed by several authors in recent literature [7,8,56,57,58,59,60,61], either as a cathode catalyst support for the oxygen reduction reaction (ORR), as a gas diffusion layer or as a component in the fabrication of polymer exchange membranes. However, the usage of CNTs as catalyst support at the anode has seldom been subject to extensive investigation [62,63,64]. The experimental evidence presented herein suggests that the functionalization of CNTs with PFPE chains reduces the overpotential for HOR, making them suitable for the fabrication of anode layers in PEMFCs. These trends are comparable to the actual state of the art materials for hydrogen storage; in particular, the overpotential of hydrogen oxidation is very low but comparable to the existing technologies [65,66,67].

The morphology of CNTs-based nanocomposites (both PE-CNT_0_ and PE-CNT_F1_) was studied by AFM, as reported in Figure 7.

The surface quality of the as-prepared samples acquired in air was significantly influenced by the sample history (sample preparation procedure, contaminants, etc.). The surface roughness was then influenced by the presence of adsorbates, clearly visible from the profiles reported at the bottom of each image. When the samples were immersed inside the electrolyte (H_2_SO_4_ 1 M), and the potential was set at the open circuit potential (OCP) value, the morphology showed significant changes (see panels 7b and 7d) with respect to the images acquired in air. In both cases, the sample prepared in air clearly showed a rough surface compared to the smoother one of the one immersed in the electrolyte. This analysis is an efficient example of the relevant effort brought by the adsorbates on the samples. The PE-CNT_0_ sample was characterized by wide swellings, while the PE-CNT_F1_ nanocomposite showed nano-structuring of the surface. The acquired profiles enhance this qualitative analysis of the two samples. In addition, we note that (at the OCP value) no further changes were observed on both PE-CNT_0_ and PE-CNT_F1_ after hours of measurement (data not reported). Reasonably, the adsorbates were dissolved when the electrochemical cell was filled, allowing the possibility to observe the electrode surface.

The morphological stability of the CNTs-nanocomposites can also be investigated during the electrochemical potential sweep of the cyclic-voltammetry. In Figure 8, we report the collected images.

Starting from the top of the images (both a and b panels), the electrochemical potential was initially set at the OCP value. From the white dot-dash line, the cyclic-voltammetry was synchronized with the AFM tip scan. Each line of the latter corresponds to a different potential value, as reported in the voltammograms placed on the left of the images. No changes were observed in the surface morphology regarding roughness or dependency on the voltage applied. Nonetheless, when we reached the cathodic current enhancement region (at about −1.0 V vs. SHE), the AFM acquisition became unstable, and it was not possible to proceed with the data acquisition. This is due to the evolution of hydrogen gas bubbles (also observed by our eyes) that perturbed the AFM cantilever. As soon as the gas bubbles were removed, the AFM acquisition was stable, and the CNTs-nanocomposites surfaces showed the same behavior.

### 3.5. Thermogravimetric Analysis

The thermogravimetric profiles of CNTs- and CB-filled nanocomposites, as well as of pristine UHMWPE, showed a degradation inflection at ~590 °C, related to the UHMWPE polymer matrix (Figure 9 and Figure 10). The samples containing carbon nanotubes showed a further signal at about 800 °C, which was ascribed to CNTs degradation (pristine CNTs thermogravimetric profile is also reported); in samples containing CB, any further inflections were observed at high temperatures due to the typical thermal stability of graphitic carbon black. The degradation of carbon nanotubes was not fully achieved in nitrogen; hence, samples containing bare or PFPE-modified CNTs showed residual weight (2–4 wt.%) at 900 °C, which is compatible with the nanofiller loading. Similarly, composites containing bare CB and PFPE-modified CB showed residues at 900 °C (3–4 wt.%).

The thermogravimetric profiles of nanocomposites containing PFPE-modified fillers showed PFPE degradation, which was expected at temperatures around 250–350 °C (Figure 9a and Figure 10a). However, the overall amount of PFPE in the nanocomposites results was close to the detection threshold of the TGA instrument (Figure 6b and Figure 7a), and differences in the decomposition onset were only observed at a low percentage of weight loss. Indeed, by comparing the thermograms of PE-CNT_0_-5, PE-CNT_F1_-5 and PE-CNT_F2_-5, it was possible to notice the slight instability of the samples containing PFPE-modified CNTs in the region between 0 and 5 wt.% of weight loss at 250–350 °C (Figure 9c). The same phenomenon occurred in PE-CB_F1_-5 (Figure 10b). This decomposition onset can be ascribed to the loss of PFPE from the nanocomposites.

## 4. Conclusions

In this work, flexible conductive composites based on ultra-high molecular weight polyethylene (UHMWPE) filled with multi-walled carbon nanotubes (CNTs) modified by perfluoropolyethers (PFPEs) were synthetized. The electrical properties, thermal stability and hydrogen evolution of UHMWPE nanocomposites filled with PFPE-modified multi-walled CNTs on nanocomposites containing CNTs were assessed by means of several different characterizations. The PFPE, added in 1:1 and 2:1 weight ratios, used as a functionalizing agent for CNTs, induced a beneficial effect on the overall conductivity of the nanocomposites, in particular at low loadings of nanofillers: the resistivity values of UHMWPE samples containing 1 wt.% of PFPE-functionalized and bare nanofillers were 2.8 × 10^3^ and 3.9 × 10^8^ Ω·m, respectively, with a significant difference of five orders of magnitude that largely overcomes the loss of conductivity. The dispersion of PFPE-functionalized CNTs in the polymeric matrix seemed to drive the formation of a continuous 3D network of CNTs characterized by enhanced conductivity and high flexibility. These preliminary results open up new promising perspectives regarding the utilization of PFPE-modified CNTs as flexible electrodes for fuel cells and hydrogen storage devices.

## Figures and Tables

**Figure 1 materials-15-06883-f001:**
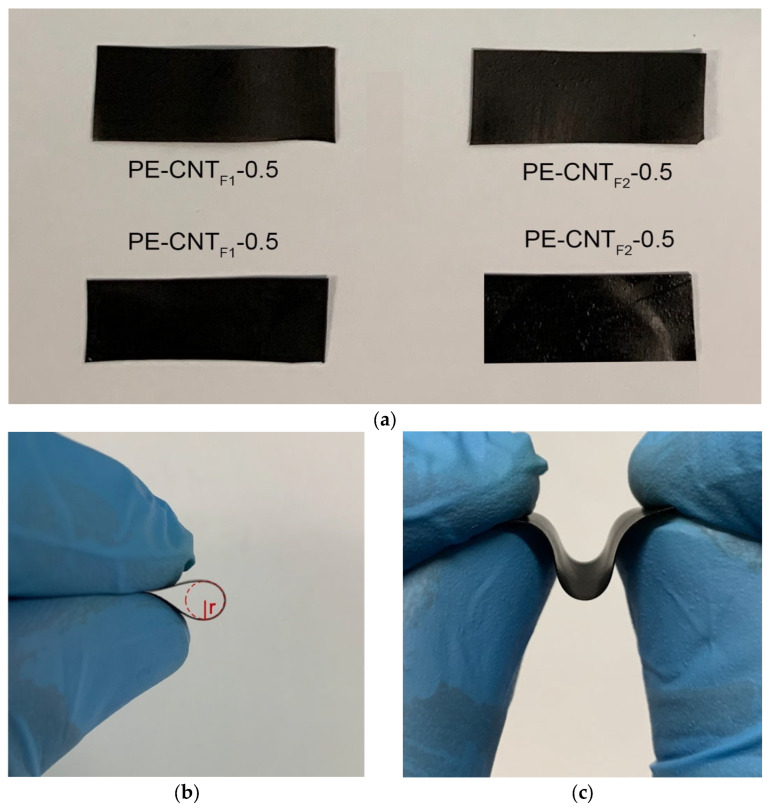
(**a**) Four samples of flexible CNT-filled ultra-high molecular weight PFPE-functionalized electrodes. In figure (**b**), the electrode is bent until reaching a bent radius of 2 mm, while in figure (**c**), the flexibility of the PE-CNT_F2_-5 electrode is displayed.

**Figure 2 materials-15-06883-f002:**
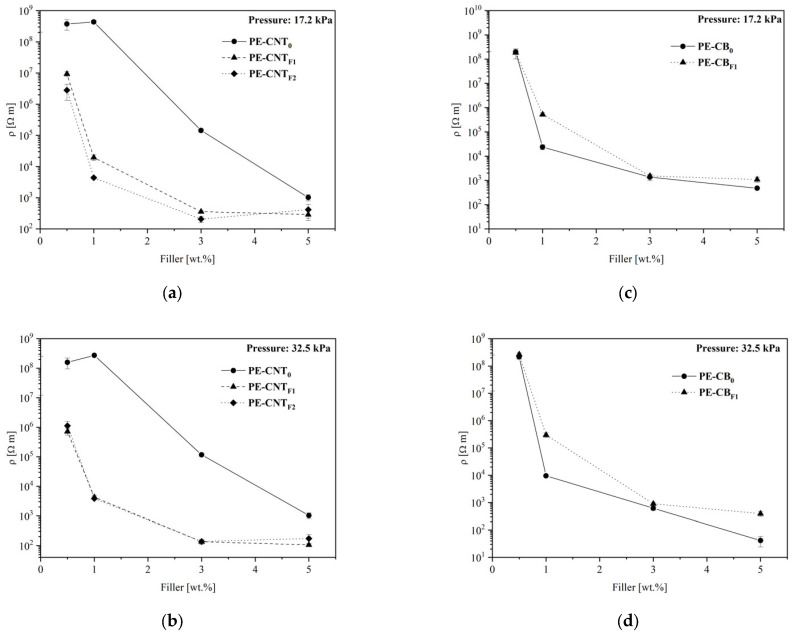
Electrical resistivity of UHMWPE-based nanocomposites as a function of the filler contents at two different applied pressures: 17.2 kPa (**a**,**c**) and 32.5 kPa (**b**,**d**). Trends referred to as PE-CNT_0_, PE-CNT_F1_ and PE-CNT_F2_ are reported in plots (**a**,**b**); those related to PE-CB_0_ and PE-CB_F1_ are shown in plots (**c**,**d**).

**Figure 3 materials-15-06883-f003:**
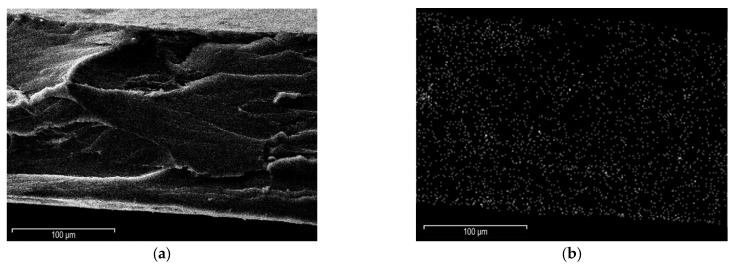
SEM micrographs of PE-CNT_F2_-3 fractured surface with a 100 k× magnification: cross-section (**a**) and elemental map (**b**), with elemental fluorine as bright spots of the analyzed sample.

**Figure 4 materials-15-06883-f004:**
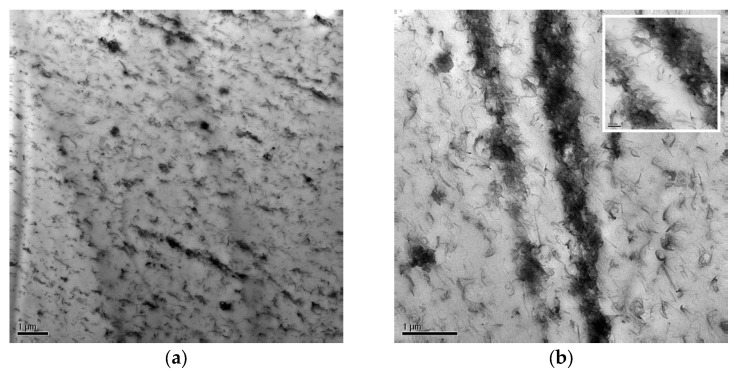
TEM images of the PFPE-modified CNTs-based UHMWPE with 1:1 ratio CNT:PFPE and 3% carbon content (PE-CNT_F2_-3). Details of the functionalized CNT spread network (**a**) and the scattered rows (**b**).

**Figure 5 materials-15-06883-f005:**
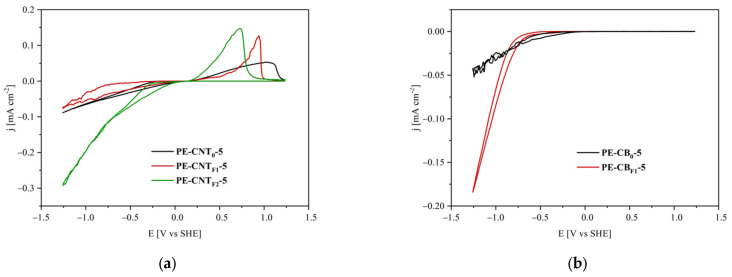
Cyclic voltammograms of PE-CNT_0_-5, PE-CNT_F1_-5, PE-CNT_F2_-5 (**a**) and PE-CB_0_-5, PE-CB_F1_-5 (**b**) nanocomposites.

**Figure 6 materials-15-06883-f006:**
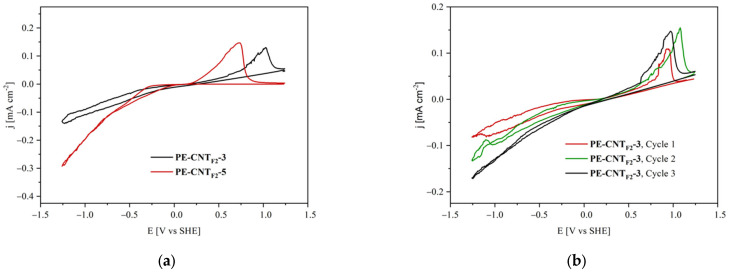
Cyclic voltammograms of PE-CNT_F2_-3 and PE-CNT_F2_-5 composites (**a**) and three consecutive runs operated on a PE-CNT_F2_-3 sample (**b**).

**Figure 7 materials-15-06883-f007:**
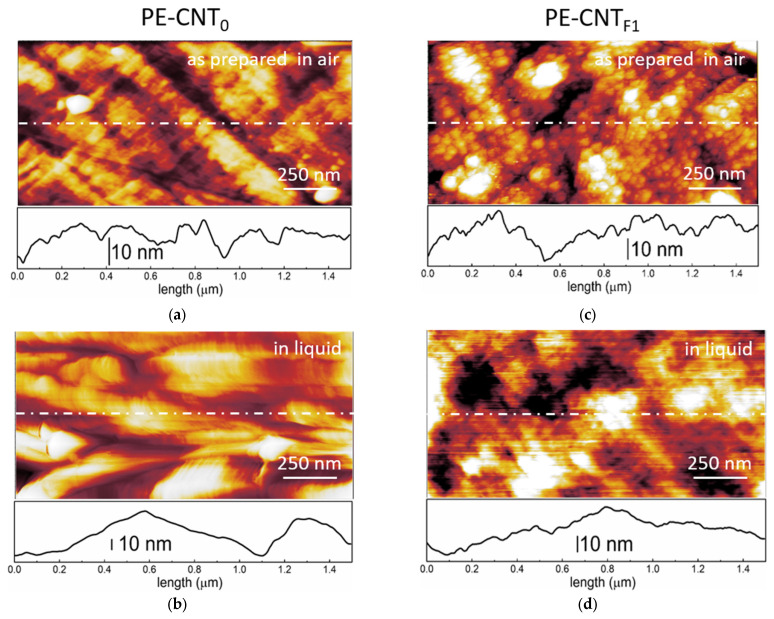
Surface morphology images [(1.50 × 0.75) μm^2^] of the CNTs-based nanocomposites. (**a**) PE-CNT_0_ sample acquired as-prepared and in air and (**b**) when the sample is immersed inside the electrolyte (H_2_SO_4_ 1 M) at the open circuit potential (OCP) (about 0.06 V vs. SHE). (**c**) PE-CNT_F1_ sample acquired as prepared and in air and (**d**) when the sample is immersed inside the electrolyte (H_2_SO_4_ 1 M) at the OCP (about 0.04 V vs. SHE).

**Figure 8 materials-15-06883-f008:**
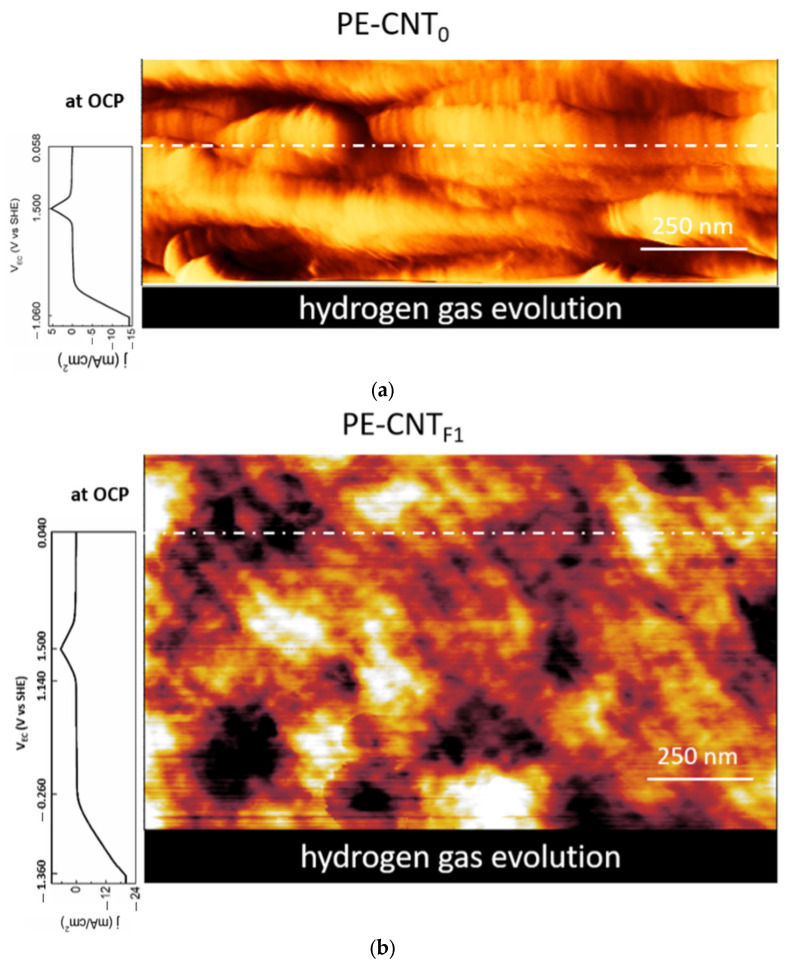
Surface morphology images of the CNTs-based nanocomposites immersed inside the electrolyte (H_2_SO_4_ 1 M) and synchronized with the cyclic-voltammetry. (**a**) PE-CNT_0_ [(1.50 × 0.55) μm^2^] and (**b**) PE-CNT_F1_ [(1.50 × 0.88) μm^2^] sample. The (**a**,**b**) scanned areas have dissimilar sizes due to different scan rates (1.5 nm/s and 3.0 nm/s, respectively) used to optimize the image quality during the acquisition. The cyclic-voltammetry scan rate is 10 mV/s for both images.

**Figure 9 materials-15-06883-f009:**
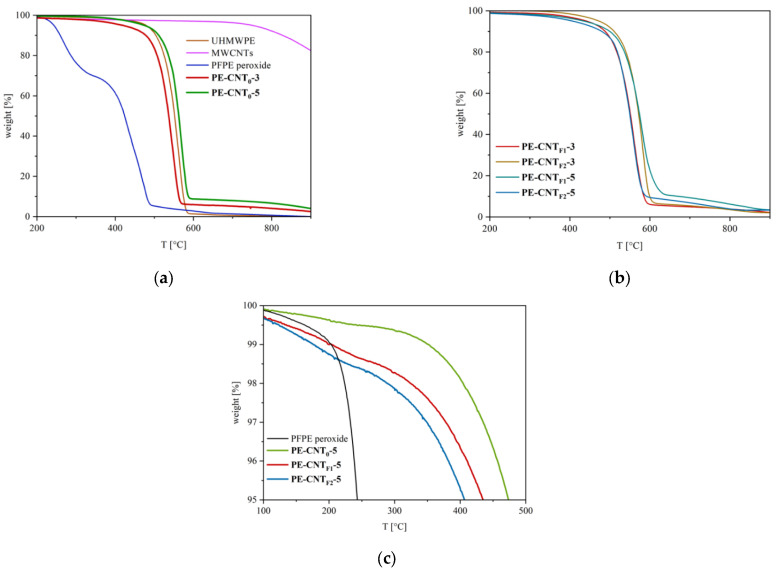
Thermogravimetric profiles of pristine UHMWPE, bare CNTs, PFPE peroxide, PE-CNT_0_-3 and PE-CNT_0_-5 (**a**), PE-CNT_F1_-3, PE-CNT_F2_-3, PE-CNT_F1_-5 and PE-CNT_F2_-5 (**b**) and a detail of the thermograms between 0 to 5 wt.% weight loss of PFPE peroxide, PE-CNT_0_-5, PE-CNT_F1_-5 and PE-CNT_F2_-5 (**c**).

**Figure 10 materials-15-06883-f010:**
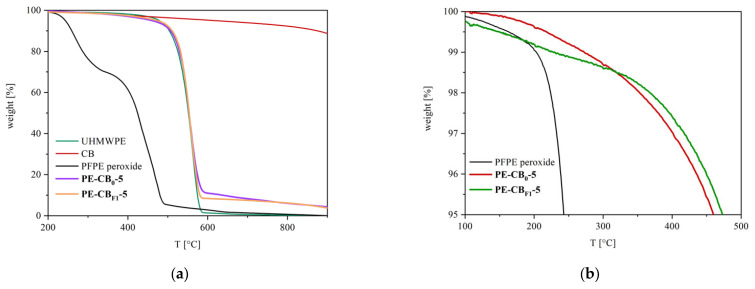
Thermograms of pristine UHMWPE, bare CB, PFPE peroxide, PE-CB_0_-5 and PE-CB_F1_-5 (**a**) and a detail of the thermograms between 0 to 5 wt.% weight loss of PFPE peroxide, PE-CB_0_-5 and PE-CB_F1_-5 (**b**).

## Data Availability

Data is contained within the article or Appendix A.

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
