# Peer review of "Flexible Perfluoropolyethers-Functionalized CNTs-Based UHMWPE Composites: A Study on Hydrogen Evolution, Conductivity and Thermal Stability"

_materials, 2022, doi:10.3390/ma15196883_

Round 1

Reviewer 1 Report

In the uploaded pdf version of the manuscript with "REVIEW" added to the file name, I have used yellow highlights and balloon comments to aid the authors in making minor grammatical corrections, correcting the numbering of some misnumbered figures, as well as to pose a few technical questions for the authors to consider.  Overall, I find this manuscript a valuable continuation of studies on such composites that the authors began studying a few years ago.  I have asked and fully expect these researchers to study and report to the scientific community the very interesting potential applications suggested by the new findings such as the use of these composites with PFPE filler as anode materials in hydrogen PEM fuel cells as well as for hydrogen storage materials.  I recommend publication after minor revision.

Reviewer 2 Report

Article entitled “Flexible perfluoropolyethers-functionalized CNTs-based UHMWPE composites: a study on hydrogen evolution, conductivity and thermal stability” contains novel findings UHMWPE composites, and use of CNTs.  However, I think that this manuscript requires major improvements in following areas:

  • Manuscript is not as per the journal template.
  • Abstract is very generalized. Authors should add major findings with quantitative result data in the abstract.  
  • Introduction is very short. I didn’t observed a single literature work in the introduction section. Add the literature of recent work carried out by the researchers and clearly mention the research gap through that study.
  • Line 49, 52, 56, and 59: Lot of bulk citations. Authors must avoid this.
  • Mention the characteristics of CNTs as well in the introduction section. Following articles will be useful for the same: https://doi.org/10.1016/j.jmrt.2021.09.038; https://doi.org/10.1016/j.matpr.2019.10.035.
  • Elaborate the research gap of this study. What was the objectives of your previous studies? Justify the necessity of current study in introduction section.
  • Lot of abbreviations were used in the present study. Describe them in their first appearance throughout the manuscript.
  • Section 2: Rename it as Materials and Methods
  • Include SEM and TEM images of the prepared CNTs in section 2.1 (if possible)
  • Add scale bar in figure 3b
  • Why the functionalized nanotubes were segregated on the surface?
  • Why only Carbon black containing composites were selected for comparison with existing nanocomposites?
  • Conclusion: First, add the brief summary in 2-3 sentences for the work carried out in present study and then start with the obtained results with bullet points.

Reviewer 3 Report

The paper "Flexible perfluoropolyethers-functionalized CNTs based UHMWPE composites: a study on hydrogen evolution, conductivity and thermal stability" described the effects of funcitionalized CNTs on the properties of UHMWPE. The topic is interesting. But this manuscript has many flaws, it should be carefully revised before publish on this journal. The detail comments are as follows:

1. The abstract should be rewritten. No experimental data and corresponding explaination was listed in this part.

2. The references are out of data. Most of them are ten years ago. The authors should cite newly references and well summarized the research background and other works.

3. The quality of SEM images in Figure 3 are boring. Please update the images, and state clearly what element in Figure 3b. In addition, scale bar is also missing in Figure 3b.

4. Please adjust the size of TEM images in Figure 4.

5. The quality of images in Figure 5 is poor. I cannot observe these curves clearly.

6. The AFM images of other samples should be compared.

7. Can you upload colorful images of the TGA curves?

8. I suggest the authors compared the functionalized CNT with prisitine CNT to prove the meaning of this work.

9. The authors should systematically compare the samples so that the readers can find the difference.

10. There are many grammar mistakes in the whole manuscript. Please revise them carefully.

Round 2

Reviewer 2 Report

Author's has somehow incorporated few comments from previous round. However most of the comments were not addressed and author's just wrote the cover letter stating the comments were incorporated. Due to lack of poor introduction section and improper justification of comments from previous round, the article in present form is not suitable for publication. 

Reviewer 3 Report

The authors have addressed the comments well. I recommend it to publish on this journal.

Round 3

Reviewer 2 Report

Accept in current form